# Systematic Review of Peer-Reviewed Literature on Global Condom Promotion Programs

**DOI:** 10.3390/ijerph17072262

**Published:** 2020-03-27

**Authors:** William D. Evans, Alec Ulasevich, Megan Hatheway, Bidia Deperthes

**Affiliations:** 1Milken Institute School of Public Health and The George Washington University, Washington, DC 20052, USA; mjhatheway@gwmail.gwu.edu; 2Independent Consultant, Silver Spring, MD 20902, USA; alec.ulasevich@gmail.com; 3United Nations Family Planning Agency, New York, NY 10017, USA; deperthes@unfpa.org

**Keywords:** HIV/STI, condoms, promotion, communication, social marketing

## Abstract

*Background*: Globally, 1.7 million people were newly infected with HIV in 2018. Condoms are inexpensive, cost-effective, reduce HIV/STI incidence, morbidity, mortality, and unintended pregnancies, and result in health care cost savings. Given the rapid increase in at-risk adolescent and young adult (AYA) populations in countries with high HIV/STI prevalence as well as the reductions in donor support, promoting consistent condom use remains crucial. We synthesized all peer-reviewed literature on condom promotion programs with a focus on promotion in low and lower middle income (LMIC) countries and with AYA users. *Methods*: We systematically reviewed the published literature. Following Preferred Reporting Items for Systematic Reviews and Meta-Analyses (PRISMA) methods, we identified 99 articles published between 2000–2019. *Results*: Condom promotion programs were generally effective in changing attitudes, social norms, and beliefs in favor of condom use, and 85% demonstrated positive effects on multiple condom use measures. Programs targeting AYA were at least equally as effective as those targeting others and often showed greater use of best practices, such as mass media (66%) and audience segmentation (31%). We also saw differences between programs in the intervention strategies they used and found greater effects of marketing strategies on AYA compared to the overall sample. *Conclusion*: Condoms remain essential to prevention, and donor support must be maintained to combat the HIV/STI epidemic.

## 1. Introduction

Despite gains made in the prevention of HIV and the scaling up of treatment programs, incidence is still alarmingly high in some settings. The recent Evidence for Contraceptive Options and HIV Outcomes (ECHO) study, for example, found an incidence rate of 3.8% among young women in select study sites [1]. Globally, an estimated 1.7 million people were newly infected in 2018, with an unmet need for contraception of some 214 million individuals and 357 million cases of curable sexually transmitted infections (STIs) [2]. While data (up to 2016) show slow but steady progress toward higher levels of condom use among non-marital, non-cohabitating men and sex workers, countries fall short of global targets (some by a substantial amount), inequities remain, and condom use in younger populations shows signs of stagnation or decline in at least a few key countries [3]. Donor funding for condom programs is also a concern; global investments in HIV prevention have declined 44% since 2012 [4]. Intensified efforts are urgently needed to increase condom use, especially in high HIV/STI prevalence countries. 

Condoms are inexpensive and cost-effective. They reduce HIV and STI incidence, morbidity, mortality, and unintended pregnancies and result in cost savings for healthcare and social sectors. Condoms are highly effective in preventing sexual transmission of HIV. Consistent and correct use of the male condom significantly reduces HIV transmission during vaginal sex (80%) [5] and anal sex (70–90%) [6,7]. Female condoms can provide similar levels of protection for vaginal sex and anal sex (although less data are available on their use during anal sex), making them among the most effective prevention technologies available today [8].

Condoms are also a familiar and convenient prevention method to most people and are, for many, still the only viable option to prevent HIV, STIs, and unintended pregnancies. Condoms are a user-initiated method, are easy to use and store, do not require medical prescriptions or direct provision by health-care personnel or in facilities, and can be used by anyone who is sexually active—including youth. Condom programming is one of five core UNAIDS Prevention Pillars [9] and should be an integrated component of all HIV prevention and care packages, offering individuals at risk with an important and effective choice to prevent HIV, STIs, and unintended pregnancies.

The current study has two overarching aims: (1) to analyze the literature in order to demonstrate which condom promotion programs have been shown to be effective and their characteristics; (2) to provide an overall description of published condom programs worldwide. We address these aims in the context of the current environment for condom demand creation and donor-sponsored programs.

Strong condom programming requires investment at the systems level in program stewardship, which will support improvements in demand and supply, leading to improved program outcomes. Mann Global Health, in close collaboration with the United Nations Family Planning Agency (UNFPA), UNAIDS, and the Bill and Melinda Gates Foundation (BMGF), has developed a theory of change for this systems level approach, as shown in Figure 1 [10].

Program stewardship is a critical function for every nationally-owned condom program to develop and implement strategies to increase use sustainably. The components of program stewardship—leadership and coordination that stretch across all sectors engaged in condom programming (public, non-profit organizations (NGO), and commercial); production and dissemination of program analytics to inform intervention design and monitor progress; financing to support needed interventions; and a supportive policy and regulatory environment—are all prerequisites for success. In particular, investments in strengthening leadership and coordination and program analytics are urgently needed in many countries. These investments at the systems level will improve quantification of need and understanding of existing use and create a strong evidence base to understand supply and demand dynamics that influence uptake and use [11]. National condom programs also require efforts to develop a supportive environment, including improved coordination and advocacy in support of a total market approach (TMA) that engages all sectors and enabling policy and regulatory environments that support diversified markets to ensure condom access is sustained. Additionally, the role of pre-exposure prophylaxis (PrEP) and the possibility that it is associated with reduced condom use, potentially leading to increased STI rates, needs to be explored as part of comprehensive condom promotion policies [12].

Given the landscape of current condom programming, there is a need to understand the state of evidence for condoms as an HIV/STI prevention strategy and an effective contraceptive method, especially for young people, as well as gaps in the knowledge base and the potential for future programs and research to move the field forward. In particular, the use of a total market approach (TMA) (i.e., creating a healthy market in which consumers who are able to pay purchase condoms, and efficient distribution ensures the poor have access to free product) is critical to the future of condom marketing [13,14].

The need for effective application of TMA is even greater given the rapid growth in the population of adolescents and young adults (AYA) in regions most affected by HIV/STI, including sub-Saharan Africa (SSA). AYA populations in SSA are estimated to increase to some 250 million under age 25 by 2030, and, given their potential sexual risk taking behavior, the HIV/STI epidemic remains a public health priority calling for comprehensive and innovative solutions [15].

Thus, there is a need to document and systematically review the state of evidence on condom programs with a specific focus on effectiveness of interventions to promote condom use through communication and social marketing, which have been the primary demand creation strategies used to date. The current research provides evidence and a framework for future programing to support TMA programs.

Overall goals of this study are to systematically review all published literature on outcomes of condom distribution programs worldwide. One important part of this project is to understand the demand generation strategies in low and lower middle income (LMIC) countries and to understand the AYA generation of condom users in order to develop effective demand promotion strategies. We hypothesize (H1) that demand generation efforts have been effective in increasing determinants of condom use, and (H2) that demand generation has been effective in increasing sales and distribution of condoms. We also ask the research question (RQ1) of whether these efforts have been more effective among specific population groups, in specific geographic locations or settings, and based on specific media channels and strategies.

In particular, we ask whether these efforts have been more effective among AYA ages 15–34 during the period in the which the project/study was conducted. Given the rapid growth of the AYA population in regions such as SSA, understanding how best to promote condoms to this priority population is critical to future prevention efforts.

## 2. Methods

We conducted a systematic search of the published, peer-reviewed literature using all relevant major online research literature databases (specified below) and following widely accepted methods for systematic review (Higgins and Green, 2011) [16]. We note that social marketing and interventions focused on condoms are also widely represented in unpublished reports and other “gray” literature. However, in this study, we focused on peer-reviewed literature in order to ensure quality of evidence and consistency with accepted systematic review practices.

We identified as relevant any manuscripts published in the English language in health, social science, and business literature that used at least one of the four Ps of marketing, had a behavioral objective targeting promotion of condom sales, use, and related behaviors, and had a health objective targeting HIV/STI or related disease prevention. We based the review methodology in part on methodologies from a previous review of branded social marketing campaigns conducted by the lead author [17]. Specifically, we searched the following health, social science, and business databases: PubMed, PsycINFO, Web of Science (includes Science Citation Index Expanded, Social Sciences Citation Index, and Arts and Humanities Citation Index), Communication & Mass Media Complete, Academic Search Premier, Business Source Premier, CINAHL, Health Source: Nursing/Academic Edition, and Health Source: Consumer Edition.

We selected search terms based on the authors’ experiences in the field and conducting previous reviews and in consultation with a medical research librarian. We applied the following criteria to conduct the search: (1) limited to only include articles published from 2000 onward; (2) search terms included condoms + marketing, brands, branding, health promotion, free distribution, commercial, subsidized; (3) went beyond other recent reviews (e.g., Evans et al., 2015) to include distribution programs for commercial brands (to extent any published results) [17]; coding included population targeted, marketing methods, research/evaluation methods, outcomes (including differential effects on audiences), type of condom (male/female), country/region, urban/peri-urban/rural, and age range target (adolescents, young adults, older).

For completeness, we also searched literature known to the authors, including publications on condom promotion, social marketing, and related intervention studies in low and middle income countries. In particular, the bibliographies of two recent meta-analyses on social marketing and mass media intervention were reviewed, and potential citations were screened following the methods described [18,19,20].

We searched all sources listed above in the date range of January 2000 to December 2019. Based on this process, we created a database of all 4240 unduplicated articles on social marketing interventions regarding condoms. Based on abstract review, we immediately excluded 3988 articles that did not relate to condom promotion or marketing, were clearly not original research, or were only focused on other family planning methods, leaving 252.

Next, we obtained and reviewed the 252 full-text articles based on our specific criteria for inclusion in study. Namely, we screened them for reports on interventions that: (1) were original research (not review papers, meta-analyses, or commentaries); (2) utilized some form of social marketing principles (i.e., reported on use of one or more of the four Ps); (3) targeted behavior relating to condom use; and (4) targeted HIV/STI prevention and/or related reproductive health outcomes as a health objective. We also screened to ensure the articles included specific reports of evaluation or implementation of social marketing activities, defined as coordinated efforts to promote, distribute, or use pricing strategies to encourage adoption of various products aimed primarily at HIV/STI prevention. Based on this in-depth screening process, we excluded 153 articles and identified 99 articles for inclusion in the study. Figure 2 summarizes the review process based on Preferred Reporting Items for Systematic Reviews and Meta-Analyses (PRISMA) methodology [21]. In this review, we followed the complete 27-item PRISMA checklist previously published [21].

Note that, due to the diverse nature of the literature on demand creation and marketing interventions in this area and the varying methods of reporting outcomes, we chose not to attempt a meta-analysis of effects of reviewed interventions on behavior. Rather, the purpose of this paper is to describe the nature of the interventions and literature, hopefully promoting more uniform reporting of condom promotion outcomes in the future.

Once the review sample of articles was identified, the first two authors individually read each of the articles in-depth and coded them for specific content reported in the results section. The results of all reviews were compiled and discussed by the reviewers. Potential sources of differences in assumptions and approaches in coding articles were identified and discussed. A consensus was reached about coding, and common procedures were adopted where discrepancies were identified.

## 3. Results

Table 1 provides a summary of basic information gleaned from each condom promotion article reviewed. The articles dealt with interventions relating to social marketing and promotion of condom brands, HIV/STI prevention programs based on increasing condom use, and family planning programs that promoted condoms. The majority of studies (81%) were focused only on male condoms, but 19% included female condoms in some fashion.

Fifty-five percent (55%) of studies were published before 2010. The majority of studies were conducted in low and middle income countries, mostly in South Asia and Sub Saharan Africa (54), with some reported in wealthy countries, mainly the USA (27). Finally, the largest single age range targeted was AYA age 15–34 (35%), followed by women (31%). Eleven percent of studies specifically targeted LGBTQ populations.

For each characteristic listed in the left-hand column, Table 1 shows the total for the overall sample (all publications reviewed), the subset for each category in the AYA population, and the subset for other populations, defined as those that did not specifically target AYA (i.e., either explicitly targeted those age 35 and older or reported no specific age targeting).

A higher percentage of articles in the 2000s reported targeting the AYA population (57%) compared to the overall population (45%), and the reverse was true in the 2010s (55% overall compared to 43% targeting AYA). Thirty-eight percent of efforts in Sub-Saharan African targeted the overall sample compared to 29% that targeted AYAs. However, 40% of efforts in the US/Canada targeted AYAs compared to 27% for this region in the overall sample.

As shown in Table 2, the interventions used a wide range of health communication and social marketing strategies, including mass media, interpersonal communication (IPC) through community outreach, and visits to households by health workers. High levels of awareness of the promoted health messages were reported. Among these, slightly more than half of studies reviewed (52%) reported use of mass media channels, with community outreach being the second most commonly reported technique (44%). Techniques to reach specific audiences were reported with segmentation being most common (21%), but in some cases, articles did not provide sufficient information to code for these categories.

There were some differences observed between AYA-targeted marketing and overall. Sixty-six percent of AYA marketing used mass media compared to 52% of the overall sample. The largest difference in mass media channel reported was 26% for AYA point of sale/location marketing efforts compared to 16% in the overall sample for that category. Among other marketing techniques reported, 31% of AYA efforts used audience segmentation (considered widely to be best marketing practice [22]) compared to 21% of the overall sample.

Table 3 provides a summary of the social marketing development described in the condom promotion articles reviewed. Forty-two percent (42%) of the studies described specific scientific theories used in the development of the social marketing effort used. Psychological theories were the most commonly used theories, found in 29% of the studies reviewed, followed by marketing theory with 19%. One third of the articles (33%) described intervention efforts based in formative research such as interviews and focus groups.

Higher percentages of efforts targeting AYA used theory, with 63% reporting some kind of theoretical framework, driven mainly by greater use of psychological theory (46% for AYA efforts compared to 29% for the overall sample). Fifty-one percent of AYA efforts reported formative research in the marketing effort compared to 33% in the overall sample, with higher percentages reported for the AYA efforts in each category of formative research methods.

Examination of the studies in terms of the use of the four Ps of marketing (place, price, product, and promotion) revealed that 45% used all four marketing techniques [22]. Promotion was the most commonly used technique reported by 83% of the articles, as represented by the channels shown in Table 2. The interventions that addressed place included any efforts to expand availability of condoms, such as community based sales, distribution of condoms by community health workers and health workers, as well as assuring availability of condoms in non-conventional venues such as hotels and places of entertainment. Three quarters of the studies reviewed (75%) mentioned such efforts to expand availability of condoms. Product interventions employed by 67% of the articles represented a wide range of techniques, from introducing new product such as female condom, introducing flavored condoms, improving existing products and packaging, to re-positioning condoms as birth control rather than STI prevention methods in some markets. Interventions focusing on price were reported by just under half (49%) of the studies. These included efforts to both reduce the actual cost of condoms through subsidies as well as to decrease the psychological price of using condoms with regular partners. 

Table 4 provides a summary of the study design and outcomes in the condom promotion articles reviewed. Most of the articles reviewed described studies with an observational design; the remaining studies were equally split between experimental and quasi-experimental designs. Almost all articles reported on the study sample size and sample characteristics. Multivariate analysis was used to report statistics in over half of the studies. More than half of the studies (57%) aimed to assess condom use behavioral objectives (i.e., the effort aimed to achieve such as outcome), including condom use at last sex, consistent condom use, and condom use with specific types of partners, and clearly stated these outcomes. A similar number of articles (56%) made clear statements about the assessing pre-behavioral objectives (i.e., the effort aimed to achieve such as outcome), including attitudes, social norms and beliefs about condom use, its benefits, acceptability, and personal preferences related to use. Correspondingly, 67% of the articles reported actually achieving some type of condom use behavioral outcome and 68% of the articles reported achieving some type of pre-behavioral outcome, as defined.

There were only small differences in the observed reporting of sample characteristics, research design type, and statistics between the overall and the AYA targeted efforts. However, higher percentages of AYA targeted efforts reported pre-behavioral and behavioral objectives (69% and 66%, respectively) compared to the overall sample (56% and 57% respectively). Thirteen percent of the overall sample reported targeting sales/distribution objectives compared to only 3% of AYA efforts. Similarly, higher percentages of AYA efforts reported achieving condom use behavior outcomes (80% and 71%, respectively) compared to the overall sample (68% and 67%, respectively), and again more of the overall sample targeted sales distribution (24%) than AYA efforts (14%).

Table 5 summarizes the significant findings reported by the articles reviewed, both overall and by the marketing approaches used. Overall, a higher percentage of the overall sample reported significant effects on condom awareness or positive reactions to promotions and sales/distribution (37% and 23%, respectively) compared to efforts targeting AYA (26% and 14%, respectively). However, a higher percentage of efforts targeting AYA reported achieving condom use pre-behavioral outcomes (71%) compared to the overall sample (58%).

Next, we analyzed the significant effects data by publications that reported capturing each outcome data category. Table 6 presents the success ratio of significant findings over the studies that reported measuring each specific outcome (i.e., the proportion of articles that reported significant findings for a particular outcome variable over the number of articles that assessed that variable), both overall and by specific intervention characteristics, such as media channels, use of theory, and marketing Ps used. Overall, almost all articles (96%) that assessed sales or distribution of condoms reported significant increase in this measure. Most articles assessed condom use pre-behavioral (attitudes, social norms, and beliefs about condom use) and condom use behavioral outcomes (use overall, frequency, and with specific partner types), and 85% each reported significant increases.

Analyzed as a ratio, we continued to see higher significant effects on awareness and reactions to promotions among the overall sample (71%) compared to AYA efforts (50%). However, we saw only small differences in effects for pre-behavioral outcomes and sales/distribution but a higher percentage of the overall sample demonstrating condom use outcomes (85%) compared to the AYA efforts (76%). Despite these differences, it should be noted that, with the exception of awareness/reactions effects among AYA efforts (50%), all the findings for significant effects among studies that measured specific outcomes were generally high (greater than 70% and most above 85%), indicating that, when condom use marketing targets these outcomes of interest, the efforts are mostly effective.

In terms of intervention characteristics, there were relatively few differences in terms of the intermediate outcomes. One notable difference was that 90% of interventions using mass media (51 in total) reported significant effects on attitudes, beliefs, and intentions to use condoms, and the same percentage was displayed among programs for AYA (23 in total).

In terms of behavioral outcomes, we saw a similar pattern of consistent effects between intervention categories, but the effects on AYA condom use were somewhat lower (72% compared to 86% for other populations) for those using mass media.

Awareness outcomes were consistently lower overall and lower among AYA compared to the general sample and the other populations. These results were consistent across all intervention characteristics.

Finally, the effects of condom promotion programs on sales were generally higher among AYA compared to the overall sample and other populations across most intervention characteristics. Across the marketing Ps, all AYA focused programs showed significant effects and were consistently higher than the overall sample and other populations, suggesting that marketing strategies were especially effective in encouraging condom purchases among AYA.

In a Appendix A to this article, Appendix A provides an overall summary of the 99 articles that underwent full text review (references provided at the end of this manuscript). The appendix table lists the study, the population targeted, the location, the product promoted, the marketing and intervention components applied, and the research design and significant effects observed.

## 4. Discussion

Sustained demand generation that results in repeat, intensive exposure to behavior change messages tailored to diverse user needs remains a critical need in many countries. Decreasing funding for condom programming across many countries has weakened efforts to develop and deliver behavior change interventions of the scale and the intensity necessary to overcome barriers to condom use [23]. Significantly more investment in demand generation activities ranging from branded and generic mass media to highly targeted interpersonal communication (IPC) is needed to ensure that people—especially youth—have the knowledge, the skills, and the power to use condoms correctly and consistently.

Condom programs must ensure that there are adequate condom supplies and distribution systems to meet current and future user demand. Though some countries have made progress in securing sufficient funding for condom procurement and for increasing condom distribution through the public sector, under-served areas remain. Condom stock-outs at the facility level and condom wastage remain challenges. Procurement of condoms in excess of reasonable projections for growth in demand also contributes to condom wastage in some countries [24]. Efforts need to include adequate condom and lubricant procurement and supplies, community-based distribution to priority populations, and targeted distribution of free commodities for those with greatest need, especially in rural and isolated locations. Decreased funding for social marketing programs and low interest from the commercial sector in reaching beyond high value urban markets also contribute to access gaps.

This study demonstrates that condoms continue to be widely promoted worldwide and, given the epidemiology of HIV/STIs, they remain a crucial technology for preventing transmission. In 2017, UNAIDS estimated that 36.9 million people were living with the HIV virus with 1.8 million new infections, the majority through unprotected sexual intercourse [25]. In addition, every day, one million people are infected with curable STIs, while over 500 million are already affected with HSV2 and 300 million with HPV [26]. There is an urgent and, to a significant extent, unmet need for women and girls to have access to condoms and other contraceptives. Most of these issues are more dramatic in SSA and among young and vulnerable people. Given the “youth bulge” in SSA that is projected to continue until the mid-21st century, there will eventually be more young people age 15–24 in the region than in India or China [27], and related increases over time in the 25–34-year-old young adult cohort. In part, as a result of these demographic changes, estimates suggest that condom availability and use will represent the largest share of total HIV infections avoided by 2030 [27].

The importance of condom promotion is suggested in part by the significant body of research in the field, some 99 published studies using the restrictive inclusion criteria in this study, and the pace of studies increased in the decade from 2010–2019 (54 of 99 reviewed) despite recent reductions in condom donor spending. However, it is worth noting that the percentage of studies showing a significant effect on condom use dropped from 87% in 2000–2009 to 78% in 2010–2019. The latter may reflect an increase in interest in evaluation data to support continued expenditures on campaigns and justify specific approaches and population focus; however, at the same time, the effects of other prevention and treatment efforts, such as PrEP, influence consistent condom use [12].

This review confirmed our original H1 that condom promotion studies are effective in promoting determinants and condom use. Condom promotion studies demonstrated a high degree of effectiveness in promoting condom use and related behavioral outcomes, with 85% demonstrating positive effects in these areas. It is worth noting that a smaller percentage of studies demonstrated message awareness in specific promotions (71%), indicating perhaps a need for better dosage/exposure measurement as well as sustained delivery of promotions.

Condom promotion studies that examined sales and distribution overwhelmingly reported an increase in sales or distribution (or both, 96%), confirming H2. Large scale condom promotion efforts reached widespread populations, including those most in need in low and middle income countries that face a disproportionate burden of HIV/STI infections and unmet contraceptive needs. This was true even in the most recent years (since 2010), when donor funding for condom distribution and promotion has declined.

Best practice marketing techniques were widely used in condom promotion, with the majority of studies employing multiple marketing channels and Ps from marketing theory. Additionally, a wide mix of marketing channels were used, and these were generally targeted to the media platforms most widely used in the country/region. In particular, there was widespread use of community outreach, radio, and outdoor advertising promotion. It is worth noting that relatively few studies used social media or other digital media to promote condoms, likely because they were often conducted in LMIC. However, this represents an opportunity to expand the reach and the recognition of condom promotion efforts.

In terms of RQ1 and whether condom promotion efforts have targeted or have been more effective among population groups such as AYA or in specific geographic locations and other settings, we made several findings. First, a disproportionate number (35 of 99) of condom promotion effort targeted AYA populations compared to other age sub-groups. Given that AYA are at greater risk, this is promising. However, it is worth noting that the majority of these AYA targeted efforts were in the US and Canada, and a disproportionately smaller number were in Sub-Saharan Africa, where condom promotion will be greatly needed in the coming decade and beyond.

We also observed greater use of mass media channels and greater use of behavior change theory in AYA condom promotion efforts compared to the overall sample. Thus, these efforts have potential to reach wider AYA audiences and are guided by solid conceptual foundations. We observed significant effects of AYA targeted efforts but overall not greater effects than on the overall sample. We did observe lower levels of effects on product awareness and positive reactions to promotions, but this may also reflect differences in media preferences and use. Given that we found very few efforts that utilized digital and social media, especially in the decade from 2010–2019, when such strategies become widespread in public health, this may reflect a need and an opportunity to shift condom promotion campaigns to the channels most used by AYA.

Overall, condom promotion efforts with AYA when specific outcomes of increasing positive attitudes, social norms, and beliefs as well as condom use behavior and sales/distribution were targeted were effective. This is promising, given that AYA are often hard to reach and at risk. This review suggests that condom promotion efforts are at least equally effective with AYA as with the overall sample of marketing promotions, and thus, considering the massive need for programs given rising AYA populations, such efforts deserve greater emphasis and attention.

There is a need for more research that sheds light on how best to build healthy markets (total market approach) for condoms [28,29]. More research is needed on how to successfully market condoms and encourage condom use using digital media marketing strategies that the youth and the young adult audience increasingly use in LMIC.

Despite its many strengths and new information yielded by this systematic review, the study has some limitations. First, promotion terminology is known to be difficult to identify in some cases due to inconsistent use of language in communication and marketing literature [30,31]. Second, we did not conduct a meta-analysis and thus cannot comment on the quality of actual data analysis or reporting of data in the reviewed papers. Finally, we acknowledge that there is substantial gray literature on communication and marketing campaigns in LMIC, and many condom promotion and distribution efforts are captured there and not in this study. For purposes of consistency and knowing the universe of articles to be screened, we elected to follow the PRISMA methodology and restrict our focus to peer-reviewed literature.

## 5. Conclusions

Despite advances in HIV treatments, including pre-exposure prophylaxis (PrEP), condoms remain an essential prevention method, especially in light of the tremendous demographic shifts in sub-Saharan Africa over the coming decade [15]. New priorities to ensure sufficient donor support for condom promotion programs are needed in the new environment for HIV/STI prevention. Condoms remain an essential strategy, and the evidence reviewed here shows that marketing promotions are effective in encouraging condom use behavior, sales, and distribution outcomes, and new intervention strategies utilizing digital media, the channel most widely used by priority populations such as AYA, should continue to be developed and evaluated.

## Figures and Tables

**Figure 1 ijerph-17-02262-f001:**
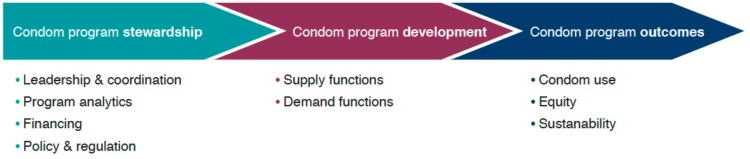
The Condom Program Pathway, a theory of change.

**Figure 2 ijerph-17-02262-f002:**
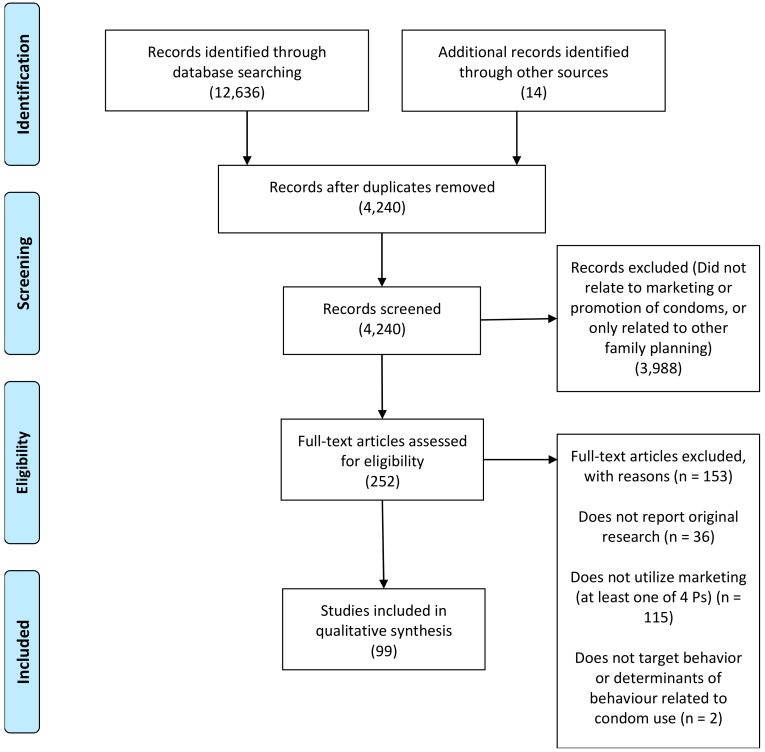
Preferred Reporting Items for Systematic Reviews and Meta-Analyses (PRISMA) diagram of systematic review process.

**Table 1 ijerph-17-02262-t001:** Characteristics of the publications reviewed.

	Overall		AYA Population		Other Populations	
**Product Promoted**	**N**	**Percent**	**N**	**Percent**	**N**	**Percent**
Male Condom	80	81%	30	86%	50	78%
Female Condom	9	9%	2	6%	7	11%
Both	10	10%	3	9%	7	11%
**Year of Publication**	**N**	**Percent**	**N**	**Percent**	**N**	**Percent**
2000 to 2009	45	45%	20	57%	25	39%
2010 to 2019	54	55%	15	43%	39	61%
**Region**	**N**	**Percent**	**N**	**Percent**	**N**	**Percent**
Sub-Sahara African	38	38%	10	29%	28	44%
US/Canada	27	27%	14	40%	13	20%
India	8	8%	0	0%	8	13%
Latin America	7	7%	2	6%	5	8%
Western Europe	5	5%	3	9%	2	3%
South East Asia	5	5%	3	9%	2	3%
East Asia	4	4%	1	3%	3	5%
India sub-continent	2	2%	0	0%	2	3%
Oceania	4	4%	4	11%	0	0%
Central Asia	1	1%	0	0%	1	2%
Middle East and North Africa	0	0%	0	0%	0	0%
Eastern/Southern Europe	0	0%	0	0%	0	0%

**Table 2 ijerph-17-02262-t002:** Marketing approaches.

Marketing Channels	Overall		AYA Population		Other Populations	
	N	Percent	N	Percent	N	Percent
Reported channels of dissemination	74	75%	25	71%	49	77%
Reported mass media channels	51	52%	23	66%	28	44%
Mass media channels mentioned						
TV	19	19%	9	26%	10	16%
Radio	33	33%	14	40%	19	30%
Print	22	22%	11	31%	11	17%
Outdoor	32	32%	13	37%	19	30%
Digital	7	7%	3	9%	4	6%
Point of Sale/Location	16	16%	9	26%	7	11%
Other	1	1%	0	0%	1	2%
Other dissemination channels						
Earned Media	2	2%	1	3%	1	2%
Community Outreach	44	44%	16	46%	28	44%
Community Mobilization	15	15%	6	17%	9	14%
Health Care Providers	25	25%	8	23%	17	27%
Mobile	2	2%	1	3%	1	2%
Social Media	4	4%	1	3%	3	5%
Other	4	4%	2	6%	2	3%
Other Marketing Techniques						
Audience Segmentation	21	21%	11	31%	10	16%
Message Tailoring	10	10%	6	17%	4	6%
Educational Entertainment	9	9%	5	14%	4	6%
Other	7	7%	3	9%	4	6%

AYA: adolescent and young adult.

**Table 3 ijerph-17-02262-t003:** Marketing development.

Coding Category	Overall		Youth and Young Adults		Other Populations	
	N	Percent	N	Percent	N	Percent
Scientific Theory Mentioned	42	42%	22	63%	20	31%
Type of Theory Mentioned						
Psychological	29	29%	16	46%	13	20%
Communication	5	5%	3	9%	2	3%
Marketing	19	19%	7	20%	12	19%
Other	1	1%	0	0%	1	2%
Formative Research Mentioned	33	33%	18	51%	15	23%
Type of Formative Research Mentioned						
Focus Groups	13	13%	8	23%	5	8%
Individual Interviews	7	7%	3	9%	4	6%
Quantitative Research	7	7%	3	9%	4	6%
Other	4	4%	4	11%	0	0%
Marketing P’s						
Product	66	67%	22	63%	44	69%
Price	49	49%	19	54%	30	47%
Place	74	75%	26	74%	48	75%
Promotion	82	83%	26	74%	56	88%

**Table 4 ijerph-17-02262-t004:** Study design and outcomes.

Coding Category	Overall		AYA Population		Other Populations	
	N	Percent	N	Percent	N	Percent
Described Sample	81	82%	28	80%	53	83%
Describe Sample Size	76	77%	25	71%	51	80%
Describe Sample Characteristics	69	70%	23	66%	46	72%
Reported Response Rate	18	18%	7	20%	11	17%
Research Design						
Not Reported	4	4%	2	6%	2	3%
Experimental	23	23%	10	29%	13	20%
Quasi-Experimental	16	16%	5	14%	11	17%
Observational	56	57%	18	51%	38	59%
Statistics Reported						
Not Reported	6	6%	4	11%	2	3%
Descriptive	35	35%	12	34%	23	36%
Multivariate	57	58%	19	54%	38	59%
Path Analysis	1	1%	0	0%	1	2%
Stated Objectives						
Product/Awareness or Reaction	42	42%	21	60%	21	33%
Condom Use Pre-Behavioral Objectives	55	56%	24	69%	31	48%
Condom Use Behavioral Objectives	56	57%	23	66%	33	52%
Sales/Distribution	13	13%	1	3%	12	19%
Outcomes Reported						
Product/Awareness or Reaction	52	53%	18	51%	34	53%
Condom Use Pre-Behavioral Outcomes	67	68%	28	80%	39	61%
Condom Use Behavioral Outcomes	66	67%	25	71%	41	64%
Sales/Distribution	24	24%	5	14%	19	30%

**Table 5 ijerph-17-02262-t005:** Significant effects reported.

Coding Category	Overall		Youth and Young Adults		Other Populations	
	N	Percent	N	Percent	N	Percent
Product/Awareness or Reaction	37	37%	9	26%	28	44%
Condom Use Pre-Behavioral Outcomes	57	58%	25	71%	32	50%
Condom Use Behavioral Outcomes	56	57%	19	54%	37	58%
Sales/Distribution	23	23%	5	14%	18	28%

**Table 6 ijerph-17-02262-t006:** Outcomes by intervention characteristics.

Intermediate outcomes by Intervention Characteristics									
**Coding category**	**Overall**			**Youth and Young Adults**			**Other Populations**		
	**N**	**Percent Reported Outcome**	**Percent Significant Effect**	**N**	**Percent Reported Outcome**	**Percent Significant Effect**	**N**	**Percent Reported Outcome**	**Percent Significant Effect**
Product Promoted									
Male Condom	80	71%	86%	30	87%	89%	50	62%	84%
Female Condom	9	56%	60%	2	50%	100%	7	57%	50%
Both	10	50%	60%	3	33%	0%	7	57%	75%
Year of Publication									
2000 to 2009	45	69%	87%	20	80%	88%	25	60%	87%
2010 to 2019	54	67%	78%	15	80%	83%	39	62%	75%
Reported Mass Media Channels	51	78%	90%	23	83%	90%	28	75%	91%
Scientific Theory Mentioned	42	86%	86%	22	91%	85%	20	80%	88%
Formative Research Mentioned	33	70%	70%	18	67%	67%	15	73%	73%
Marketing P’s									
Product	66	70%	80%	22	82%	78%	44	64%	82%
Price	49	76%	81%	19	79%	73%	30	73%	86%
Place	74	65%	85%	26	77%	80%	48	58%	89%
Promotion	82	73%	83%	26	100%	78%	56	61%	82%
Sample	99	68%	85%	35	80%	89%	64	61%	82%
**Behavioral Outcomes by Intervention Characteristic**									
**Coding category**	**Overall**			**Youth and Young Adults**			**Other Populations**		
	**N**	**Percent Reported Outcome**	**Percent Significant Effect**	**N**	**Percent Reported Outcome**	**Percent Significant Effect**	**N**	**Percent Reported Outcome**	**Percent Significant Effect**
Product Promoted									
Male Condom	80	68%	82%	30	73%	68%	50	64%	91%
Female Condom	9	67%	67%	2	100%	100%	7	57%	50%
Both	10	60%	83%	3	33%	100%	7	71%	80%
Year of Publication									
2000 to 2009	45	73%	79%	20	70%	79%	25	76%	79%
2010 to 2019	54	61%	82%	15	73%	64%	39	56%	91%
Reported Mass Media Channels	51	78%	80%	23	78%	72%	28	79%	86%
Scientific Theory Mentioned	42	81%	82%	22	82%	83%	20	80%	81%
Formative Research Mentioned	33	67%	77%	18	67%	83%	15	67%	70%
Marketing P’s									
Product	66	71%	81%	22	82%	78%	44	66%	83%
Price	49	78%	79%	19	79%	73%	30	77%	83%
Place	74	68%	84%	26	73%	79%	48	65%	87%
Promotion	82	71%	83%	26	88%	78%	56	63%	86%
Sample	99	67%	85%	35	71%	76%	64	64%	84%
**Awareness Outcomes by Intervention Characteristics**									
**Coding category**	**Overall**			**Youth and Young Adults**			**Other Populations**		
	**N**	**Percent Reported Outcome**	**Percent Significant Effect**	**N**	**Percent Reported Outcome**	**Percent Significant Effect**	**N**	**Percent Reported Outcome**	**Percent Significant Effect**
Product Promoted									
Male Condom	80	51%	68%	30	50%	47%	50	52%	81%
Female Condom	9	56%	80%	2	50%	0%	7	57%	100%
Both	10	60%	67%	3	67%	50%	7	57%	75%
Year of Publication									
2000 to 2009	45	47%	62%	20	35%	29%	25	56%	79%
2010 to 2019	54	57%	74%	15	73%	55%	39	51%	85%
Reported Mass Media Channels	51	57%	76%	23	48%	55%	28	64%	89%
Scientific Theory Mentioned	42	64%	63%	22	50%	46%	20	80%	75%
Formative Research Mentioned	33	52%	59%	18	56%	50%	15	47%	71%
Marketing P’s									
Product	66	62%	73%	22	55%	50%	44	66%	83%
Price	49	63%	74%	19	53%	50%	30	70%	86%
Place	74	57%	74%	26	54%	50%	48	58%	86%
Promotion	82	55%	71%	26	65%	47%	56	50%	86%
Sample	99	53%	71%	35	51%	50%	64	53%	82%
**Sales outcomes by Intervention Characteristics**									
**Coding category**	**Overall**			**Youth and Young Adults**			**Other Populations**		
	**N**	**Percent Reported Outcome**	**Percent Significant Effect**	**N**	**Percent Reported Outcome**	**Percent Significant Effect**	**N**	**Percent Reported Outcome**	**Percent Significant Effect**
Product Promoted									
Male Condom	80	25%	90%	30	13%	100%	50	32%	88%
Female Condom	9	0%	NA	2	0%	NA	7	0%	NA
Both	10	40%	100%	3	33%	100%	7	43%	100%
Year of Publication									
2000 to 2009	45	16%	86%	20	10%	100%	25	20%	67%
2010 to 2019	54	31%	94%	15	20%	100%	39	36%	91%
Reported Mass Media Channels	51	22%	100%	23	17%	100%	28	25%	100%
Scientific Theory Mentioned	42	12%	80%	22	5%	100%	20	20%	75%
Formative Research Mentioned	33	12%	75%	18	11%	100%	15	13%	50%
Marketing P’s									
Product	66	20%	85%	22	9%	100%	44	25%	82%
Price	49	27%	92%	19	16%	100%	30	33%	90%
Place	74	30%	91%	26	19%	100%	48	35%	88%
Promotion	82	20%	94%	26	12%	100%	56	23%	92%
Sample	99	24%	96%	35	14%	100%	64	30%	90%

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
