# Peer review of "Systematic Review of Peer-Reviewed Literature on Global Condom Promotion Programs"

_ijerph, 2020, doi:10.3390/ijerph17072262_

Round 1

Reviewer 1 Report

This manuscript decribes the systematic review of published condom promotion interventions. Overall this article provides a good review of condom promotion interventions and is useful for those in the field.

There are some problems with the manuscript.

Statements that are unclear-

The first sentence of abstract is not completed - 1.7 million new infections of what? Looks like HIV.

The paper often focuses on HIV but then will  refer to STIs and family planning. Does not really get into the assertion that use of PrEP might reduce condom use and increase STIs. Programmatically this is important for the support of continues condom use in high risk populations.

suffers from excessive content - paragraphs in lines 58-112 of background are unnecessary and should be removed, reduced and/or move to discussion.

Table 6 is very large but provides the key information of about how the interventions worked but it is hard to know what is meant by the intermediate, behavioral, awareness and sales outcomes. Note typo in titles for sales. This is an important for the interpretation of the finding. Should include was the types of outcomes were that would be counted in the main body of the paper or as a footnote to table.

Author Response

We thank the reviewer for these valuable comments and respond as follows:

  • We corrected the abstract as noted.
  • We addressed consistent use of terminology.
  • We noted that some research shows PrEP may reduce condom use, thus increasing STI risks, and added a citation.
  • We have edited and shortened the content from lines 58-112 as requested.
  • We have edited table 6 and clarified associated text as noted.

Reviewer 2 Report

Manuscript ID: ijerph-736814

Thank you for the opportunity to review this systematic review of Peer-reviewed Literature on Global Condom Promotion Programs.

This is an important and thorough review that requires little revision. However, it could benefit from consideration of a few points, listed below.

Abstract

  1. Highlighting that the number of articles has increased in the last decade compared to the previous decade, does not seem to be a major point to include, especially as there was not a great difference.

Methods

  1. Did the search include any language? Were any articles in languages other than English identified?
  2. Was ‘health promotion’ included in the search terms? Can the search terms be included in the article?
  3. Per the PRISMA method, was a quality assessment of the risk of bias in individual studies undertaken (including specification of whether this was done at the study or outcome level)? How was this information used in data synthesis?
  4. Also, per the PRISMA method, was a risk of bias across studies assessment (that may affect the cumulative evidence e.g., publication bias, selective reporting within studies)?

Discussion

  1. Page 14 of 22, Line 335, ‘studies studied…’ is an awkward expression.
  2. In relation to articles published in the last decade, compared to the previous decade, I noticed that the percentage of significant effect decreased (from 87% to 78%). This could be worthy of exploration in the discussion.

Table 1

  1. Table 1 has a significant number of typos.
  2. Are the URLs required? These could best be deleted.
  3. The articles are classified by regions. However, in Table 1 ‘Pacific Islands’ currently includes Fiji, New Zealand and Indonesia. Indonesia is part of South East Asia not Pacific Islands. The Potter et al article includes ‘Australia’ but not a region that Australia belongs to. Australia can be grouped with Fiji and New Zealand and named ‘Australasia’ or ‘Oceania’ rather than ‘Pacific Islands’.

Author Response

We thank the reviewer for these valuable comments and respond as follows:

  • We have edited the abstract as noted, but kept the small point about number of articles over time in the main narrative as it shows the consistent growth of literature in the field of interest.
  • Only articles published in English were included, and we added this point to the methods.
  • Health promotion was included and the search terms are listed in lines 130-131.
  • We did address the points made about the PRISMA method and have added text about this to the methods.
  • We made the noted revisions to the discussion, and addressed the point about the decrease in significant effects in papers published in the 2010s, with an added hypothesis tied to another reviewer's comment about the role of PrEP.
  • We made the noted revisions to Table 1.